# An Advanced Approach for MgZnAl-LDH Catalysts Synthesis Used in Claisen-Schmidt Condensation

**Rodica Zăvoianu** [1,2], **Silvana-Denisa Mihăilă** [1,2], **Bogdan Cojocaru** [1,2], **Mădălina Tudorache** [1,2], **Vasile I. Pârvulescu** [1,2], **Octavian Dumitru Pavel** [1,2,*], **Solon Oikonomopoulos** [3] and **Elisabeth Egholm Jacobsen** [3,*]

[1] Department of Organic Chemistry, Biochemistry & Catalysis, Faculty of Chemistry, University of Bucharest, 4–12 Regina Elisabeta Blv., 030018 Bucharest, Romania; rodica.zavoianu@chimie.unibuc.ro (R.Z.); silvana.mihaila@s.unibuc.ro (S.-D.M.); bogdan.cojocaru@chimie.unibuc.ro (B.C.); madalina.sandulescu@g.unibuc.ro (M.T.); vasile.parvulescu@chimie.unibuc.ro (V.I.P.)

[2] Research Center for Catalysts & Catalytic Processes, Faculty of Chemistry, University of Bucharest, 4–12 Regina Elisabeta Blv., 030018 Bucharest, Romania

[3] Department of Chemistry, Norwegian University of Science and Technology, Høgskoleringen 5, 7491 Trondheim, Norway; solon.oikonomopoulos@ntnu.no

[*] Correspondence: octavian.pavel@chimie.unibuc.ro (O.D.P.); elisabeth.e.jacobsen@ntnu.no (E.E.J.); Tel.: +40-21-305-1464 (O.D.P.); +47-98-843-559 (E.E.J.)

**Abstract:** Using organic-base tetramethylammonium hydroxide (TMAH) is a viable, cheap, and fast option for the synthesis of MgZnAl-LDH-type materials by both co-precipitation and mechano-chemical methods. TMAH presents several advantages, such as the smaller quantity of water required in the washing step compared to the use of inorganic alkalis, the prevention of LDH contamination with alkali cations, and its action as a template molecule in texture tailoring. It also has disadvantages, such as its presence in small quantities in the resulting layered materials. Regardless of the use of organic/inorganic bases and co-precipitation/mechano-chemical methods, zincite stable phase was found in all the synthesized solids. The basicity of catalysts followed the trend: mixed oxides > reconstructed > parent LDH. The memory effect of LDH was supported only by the presence of Mg and Al cations, while Zn remained as a zincite stable phase. The catalytic activities for Claisen–Schmidt condensation of benzaldehyde with cyclohexanone provided values higher than 90% after 2 h, with a total selectivity toward 2,6-dibenzylidenecyclohexanone, while self-condensation of cyclohexanone yielded no more than 7.29% after 5 h. These behaviors depended on catalyst basicity as well as on the planar rigidity of the compound.

**Keywords:** layered double hydroxides (LDH); mechano-chemical/co-precipitation synthesis; organic alkalis (tetramethylammonium hydroxide); memory effect; Claisen-Schmidt condensation; self-cyclohexanone condensation

## 1. Introduction

Recently, a remarkable increase in layered double hydroxide (LDH) utilization in various domains, for instance as catalysts for fine chemicals syntheses [1–4], adsorbents for environmental protection [5–7], corrosion inhibitors [8], polymer additives [9], drug delivery agents [10], building materials [11], in battery manufacture [12,13], etc. These materials have a general formula $[M^{2+}_{1-x}M^{3+}_x(OH)_2]^{x+}[A^{n-}_{x/n}]\cdot mH_2O$ [4], where $M^{2+}$ and $M^{3+}$ are bi- and trivalent cations adopting an octahedral geometry [14], **A** is an anion with charge **n**, **x** is equal to the ratio $M^{3+}/(M^{2+} + M^{3+})$, and **m** is the number of water molecules. This formula is not restricted to compositions with two different cations, but includes those with multiple types of bivalent and trivalent cations, which may generate ternary or quaternary compounds [15,16]. A specific property of these materials is the so-called *memory effect*, which allows the reconstruction of the layered structure by hydrating the mixed oxides obtained from thermal treatments of the parent LDH at up to 500–600 °C

[4]. This reconstruction of the layered structure is carried out by two mechanisms: *(i)* a dissolution–re-crystallization mechanism suggested by Ulibarri and Takehira [17,18] and *(ii)* retro–topotactic transformation, proposed by Marchi and Apesteguía [19]. Besides the fact that a specific basicity can be afforded by the partial replacement of the interlayer carbonate anions with some pronounced basic ones [20], i.e. hydroxyl groups, it is also possible to insert the layered structure anions with large dimensions [21]. Traditionally, the layered catalytic materials are obtained by several methods [22], such as co-precipitation (at low or high supersaturation, urea hydrolysis), ion-exchange, rehydration using structural memory effect, hydrothermal method, secondary intercalation, etc. Other methods, which include mechano-chemical synthesis [23], exfoliation in aqueous solution [24], dry exfoliation [25], and electrochemical synthesis [26], were developed recently aiming to replace the usual methods. However, most traditional methods use inorganic alkalis (i.e. NaOH, KOH, $Na_2CO_3$, $K_2CO_3$ etc.), which can often lead to LDH contamination with traces of alkaline metals cations (e.g., *ppm* or *ppb* levels). Besides these, the need for large amounts of energy or water (in the washing step) or the need for specific glassware for the preparation are other disadvantages that these inorganic alkalis generate. A new approach that removes all these drawbacks considers the use of organic alkalis for LDH synthesis [27–29]. However, the replacement of inorganic alkalis with organic ones could be somehow restrictive due to their price, inadequate solubility in water, and tendency to decompose during preparation or storage over time, as well as the possible presence of impurities in their composition, etc. However, all these issues can be avoided by proper organic alkali selection and manipulation under optimal conditions. Urea and hexamethylenetetramine can act as precipitating agents using hydrolysis under pressure and temperatures higher than 100 °C [30].

Among the possible utilizations of LDHs, an important function is their use as catalysts for fine chemical synthesis. Since 1880, when Schmidt [31] reported for the first time the aldol condensation between furfural and acetaldehyde/acetone being developed by Claisen [32], aldol condensation became one of the most important reactions in organic chemistry, where a C—C bond is generated in the presence of certain catalysts with acid/base character [3]. A special example of Claisen–Schmidt condensation is the reaction between benzaldehyde and cyclohexanone, which generates two valuable fine chemical compounds, i.e. 2-benzylidenecyclohexanone (2-BCHO) [33] and 2,6-dibenzylidenecyclohexanone (2,6-DBCHO) [34]. Which parameters are responsible for selecting the direction towards *mono-* or *di-*condensation products is a matter still under debate. Until now, Tang [35] and Sluban [36] obtained a high yield of 96% for the *mono-*condensation product 2-BCHO in the presence of modified CaO with benzyl bromide and HTiNTs. Meanwhile, there are numerous publications reporting higher yields for the *di-*condensation product (2,6-DBCHO). Thus, an activated fly ash catalyst [37] generated 56% yield; a ground solid NaOH led to a 2,6-DBCHO yield of over 96% [38]; NaP Zeolite/$CoFe_2O_4$/Am-Py (1:4) catalyst in free solvent condition [39] yielded 92% 2,6-DBCHO; using quaternary ammonium surfactants with *n*-hexadecyl group micellar system ($C_{16}QAS$) as catalyst [40] produced a yield of 73.8% for 2,6-DBCHO; and $SOCl_2$, in anhydrous ethanol ($SOCl_2$/EtOH) [41] yielded more than 92%. A common feature of all the above-mentioned catalysts is the fact that they have base sites that contribute to the activation of reagent molecules.

Nevertheless, during the Claisen–Schmidt condensation, two secondary reactions may occur [3], i.e. the cyclohexanone self-condensation to 2-(1-cyclohexenyl)-cyclohexanone (2-CyCHO) and 2,6-dicyclohexylidencyclohexanone (2,6-DCyCHO) compounds in the presence of catalytic base sites, and the oxidation of benzaldehyde to benzoic acid, especially since it is known that this reaction takes place non-catalytically under ambient conditions in the presence of air. To our knowledge, until now there have been no reports concerning the use of LDH-type solids in this reaction.

Given the above, this paper aims to bring new insights into the influence of the synthesis methods (traditional co-precipitation/non-traditional mechanical-chemical)

and alkalis used (traditional inorganic/non-traditional organic) in tailoring the physico-chemical properties of MgZn/Al LDH. In this work, we focus on using tetramethylammonium hydroxide (TMAH), an organic alkali that is cheaper than hexamethylenetetramine, is easily soluble in water, and allows the induction of precipitation at atmospheric pressure. Additionally, the optimum activity of LDH-type materials, mixed oxides, and reconstructed LDH are highlighted in the Claisen–Schmidt condensation between benzaldehyde and cyclohexanone and in the cyclohexanone self-condensation.

## 2. Results and Discussion

### 2.1. Characterization of Catalysts

The XRD pattern for the LDH synthesized through the traditional route of co-precipitation in the presence of inorganic alkalis, LDH-MgZnAl-$CO_3^{2-}$/$OH^-$-CP, presented sharp and symmetric reflections at small angles for the (003), (006), (009), (012), (015), and (018) planes, while at higher angles there were broad and weak reflections for the (110) and (113) planes, which were representative of LDH-type materials (ICDD 70-2151), Figure 1A [23].

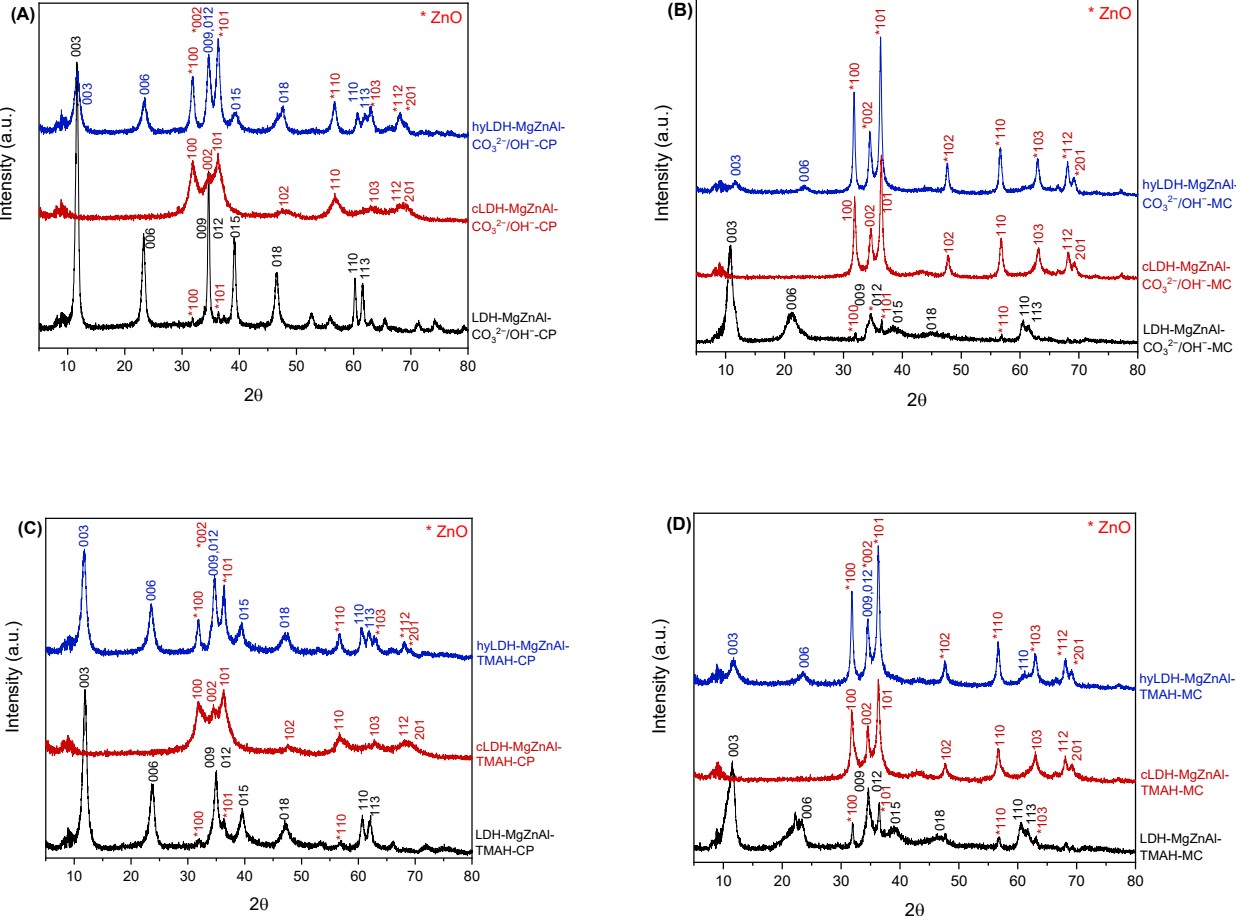

**Figure 1.** XRD patterns of the materials synthesized though (**A**) co-precipitation and (**B**) mechano-chemical method, both in the presence of $Na_2CO_3$/NaOH, and (**C**) co-precipitation and (**D**) mechano-chemical method, both in the presence of TMAH; (* - ZnO).



These reflections were indexed in a hexagonal lattice with an *R3m* rhombohedral symmetry. Aside from these, some additional fine diffraction lines corresponding to zincite phase appeared in the domain of 2θ = 31–38° (ICDD 005-0664). Its presence was noticeable in the XRD pattern of the calcined material, cLDH-MgZnAl-$CO_3^{2-}$/$OH^-$-CP, where diffraction lines corresponding to ZnO ((100), (002), and (101)) were more intense. Reconstruction by memory effect led to a mixture of stable zincite phase and Mg/Al LDH. At the same time, the IFS parameter (i.e., the interlayer distance) decreased from 2.82 Å to 2.80 Å, and 2θ_{003} shifted towards a higher value from 11.6259° to 11.6421°, as shown in Table 1, thus denoting the presence of smaller species in the interlayer space, which, according to the literature [20], are $OH^-$ groups that partially replace the $CO_3^{2-}$ groups following the calcination–reconstruction process. This behavior was also observed in the DRIFT spectra, as shown in Figure 2.

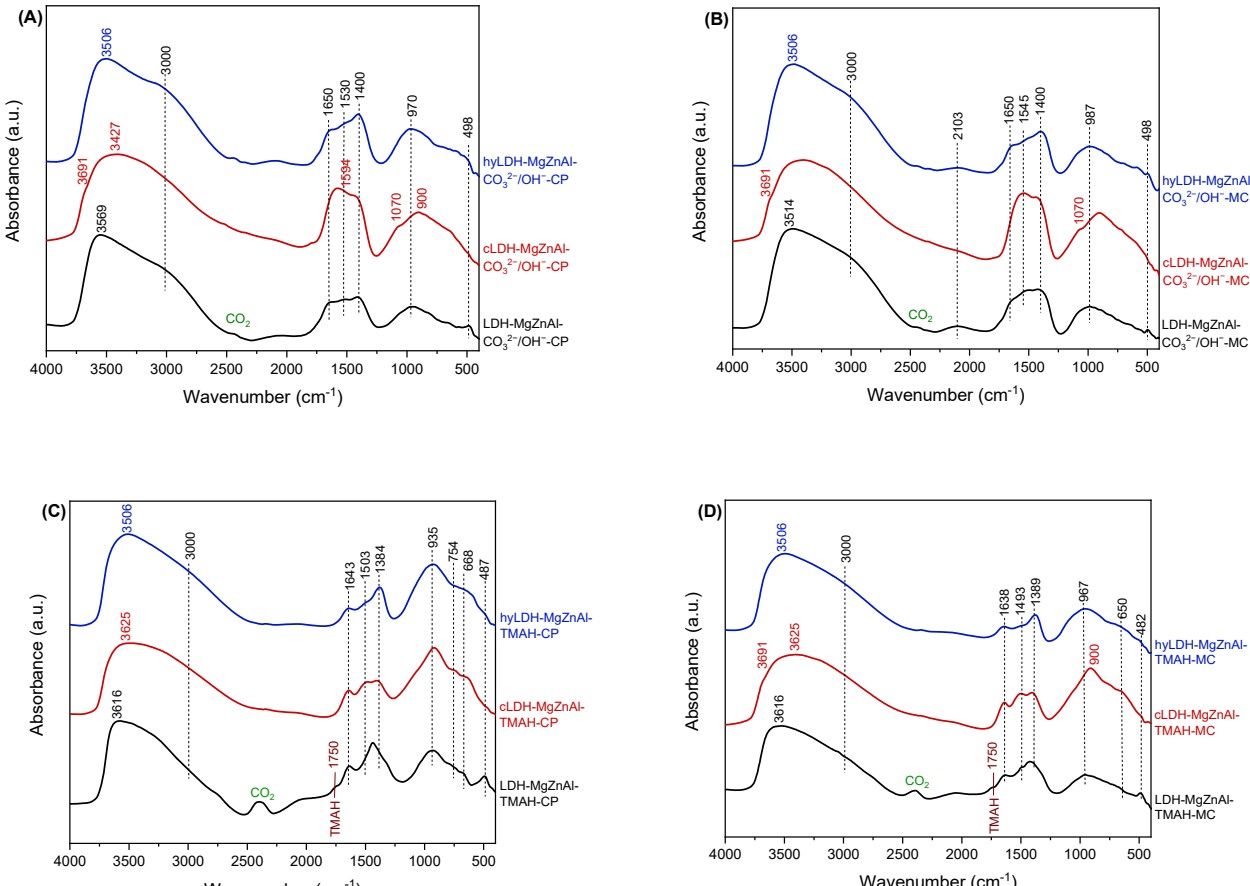

**Figure 2.** DRIFT spectra of the materials synthesized though (**A**) co-precipitation and (**B**) mechano-chemical method, both in the presence of $Na_2CO_3$/NaOH, as well as (**C**) co-precipitation and (**D**) mechano-chemical method, both in the presence of TMAH.

**Table 1.** The network parameter of samples.

| Hydrotalcite Samples | Lattice Parameters | | IFS * (Å) | 2θ_{003} (°) | I_{003}/I_{006} | I_{003}/I_{110} | FWHM_{003} | D ** (Å) |
|---|---|---|---|---|---|---|---|---|
| | *a* (Å) | *c* (Å) | | | | | | |
| LDH-MgZnAl-$CO_3^{2-}$/$OH^-$-CP | 3.0717 | 22.8689 | 2.82 | 11.6259 | 2.96 | 5.54 | 0.6064 | 131.8 |
| hyLDH-MgZnAl-$CO_3^{2-}$/$OH^-$-CP | 3.0497 | 22.8017 | 2.80 | 11.6421 | 1.97 | 3.42 | 1.1043 | 72.4 |
| LDH-MgZnAl-$CO_3^{2-}$/$OH^-$-MC | 3.0504 | 22.4314 | 2.68 | 11.8652 | 2.54 | 5.33 | 0.8885 | 90.0 |

| Sample | $a$ (Å) | | | | | | FWHM | D (Å) |
|---|---|---|---|---|---|---|---|---|
| hyLDH-MgZnAl-CO$_3^{2-}$/OH$^-$-MC | 3.0497 | 22.6427 | 2.75 | 11.7234 | 2.20 | 4.48 | 0.9709 | 82.3 |
| LDH-MgZnAl-TMAH-CP | 3.0617 | 24.7662 | 3.46 | 10.8279 | 3.57 | 5.08 | 1.1644 | 68.6 |
| hyLDH-MgZnAl-TMAH-CP | 2.9757 | 22.7965 | 2.80 | 11.6610 | 1.59 | 1.13 | 0.9020 | 88.6 |
| LDH-MgZnAl-TMAH-MC | 3.0580 | 23.6969 | 3.10 | 11.3150 | 2.83 | 3.45 | 1.6050 | 49.8 |
| hyLDH-MgZnAl-TMAH-MC | 3.0535 | 22.7515 | 2.78 | 11.6700 | 2.18 | 3.45 | 1.1800 | 67.7 |
| **Mixed oxides samples** | **$a$ (Å)** | | | **$2\theta_{101}$ (°)** | **$I_{003}$** | | **FWHM$_{101}$** | **D *** (Å)** |
| cLDH-MgZnAl-CO$_3^{2-}$/OH$^-$-CP | 4.9536 | | | 36.2400 | 279 | | 1.8266 | 45.8 |
| cLDH-MgZnAl-CO$_3^{2-}$/OH$^-$-MC | 4.9483 | | | 36.3628 | 257 | | 1.5500 | 53.9 |
| cLDH-MgZnAl-TMAH-CP | 4.9332 | | | 36.3949 | 736 | | 0.4925 | 169.9 |
| cLDH-MgZnAl-TMAH-MC | 4.9449 | | | 36.3055 | 175 | | 0.6023 | 138.9 |

* IFS represents the interlayer free distance; 4.8Å brucite sheet thickness [42]. ** D represents the mean crystallite size (derived from the Debye–Scherrer equation) determined from the FWHM of the (003) reflection for LDH samples. *** D represents the mean crystallite size (derived from the Debye–Scherrer equation) determined from the FWHM of the (101) reflection for mixed oxides.

The easily noticeable decrease in $a$ network parameter (i.e. the distance between the network cations) was due to the extraction of Zn from the LDH network followed by the stable phase of zincite synthesis made by calcination. The mechano-chemical LDH, LDH-MgZnAl-CO$_3^{2-}$/OH$^-$-MC, presented particularities similar to the above-mentioned, as shown in Figure 1B, but the amount of zincite increased and that of the LDH phase decreased. Moreover, the number of interlayer species with small size, i.e. OH$^-$, is higher compared to LDH obtained by co-precipitation, a fact highlighted by an IFS value of 2.68 Å compared to 2.82 Å, and a shifting from $2\theta_{003}$ to 11.8652°. After the reconstruction of the layered structure hyLDH-MgZnAl-CO$_3^{2-}$/OH$^-$-MC, a large amount of zincite remained as the stable phase. This behavior simultaneously led to an increase in the IFS value of 2.75 Å, as well as a shift to lower values of 11.7234 for $2\theta_{003}$. The mean crystallite size derived from the Debye–Scherrer equation shows an insignificant variation (from 90.0 Å to 82.3 Å) compared to that noticed for co-precipitated materials (from 131.8 Å to 72.4 Å).

The use of TMAH in LDH synthesis by both co-precipitation and mechano-chemical methods produced a pure LDH structure mixed with small amounts of zincite phase, as shown in Figure 1C,D. The IFS values of LDH obtained by both methods were increased (LDH-MgZnAl-TMAH-CP of 3.46 Å and LDH-MgZnAl-TMAH-MC of 3.10 Å), because, besides OH$^-$ and CO$_3^{2-}$ groups, in the interlayer space there were also small amounts of TMAH as well as tri-methyl amine (a ubiquitous impurity from TMAH; its existence being also demonstrated by DRIFT). The calcination processes eliminated both organic compounds as dimethyl ether and methanol [43]. The LDH structure reconstruction through the memory effect was present for all materials synthesized in the presence of TMAH. The diffraction lines of hyLDH-MgZnAl-TMAH-CP and hyLDH-MgZnAl-TMAH-MC were more intense compared to those of the materials synthesized with inorganic alkalis. All mixed oxide samples prepared by calcination of parent LDH showed only ZnO lines with no important differences in $a$ network parameter and $2\theta_{101}$, except for the mean crystallite size, where organic alkalis generated values three times higher compared with the inorganic one, as shown in Table 1. Because the diffraction lines corresponding to the Mg(Al$^{3+}$)O solid solution of MgO-periclase-type (JCPDS-45-0946) phases were not obvious, this phase was highly dispersed.

DRIFT spectra of the investigated LDH, as shown in Figure 2A–D, presented a large band in the 3700–3400 cm$^{-1}$ domain corresponding to the vibration of hydroxyl groups, $\upsilon_{O-H}$, which at 3000 cm$^{-1}$ was assigned to hydrogen bonds between carbonate anion and water molecules, both situated in the interlayer space [44], a band at 1638–1650 cm$^{-1}$ characteristic of the H$_2$O bending vibration of interlayer LDH structure, and a band at 1200–600 cm$^{-1}$ assigned to the CO$_3^{2-}$ group vibration, while that below 600 cm$^{-1}$ was assigned to Mg–O, Zn–O, and Al–O bonds.

The band at 1750 cm$^{-1}$ for samples synthesized with TMAH was assigned to this hydrolysis agent, which was present in small quantities in the pores of LDH. This remnant compound, due to its pronounced base character, generated a higher physisorption of atmospheric $CO_2$, exhibiting a band at 2450 cm$^{-1}$ (stronger than that appearing in the spectra of materials prepared with inorganic alkalis). It is noteworthy that in the case of the TMAH prepared samples, the bands from 1650 cm$^{-1}$ and 1400 cm$^{-1}$ shifted to 1640–1638 cm$^{-1}$ and 1389–1384 cm$^{-1}$, respectively. The calcination at 460 °C eliminated the remnant TMAH and adsorbed $CO_2$, leading to the total disappearance of their corresponding IR adsorption bands. In addition, there was also a partial removal of $OH^-$ and $CO_3^{2-}$ from the network, leading to a decreased intensity of the corresponding bands, as also remarked by other authors working on LDH prepared with inorganic alkalis, even when the calcination was performed at 650–700 °C [19]. The spectra of the calcined samples also present a band of characteristic stretching vibrations of structural hydroxyl groups, which were coordinated to Mg or Al octahedral at 3691 cm$^{-1}$ [45]. This phase was evidently highly dispersed in the solid matrix since its reflections were absent in the XRD patterns. The bands at 3000 cm$^{-1}$ and that at 1638–1650 cm$^{-1}$ were well restored in the spectra of the reconstructed samples. Simultaneously, the band at 1400–1384 cm$^{-1}$ became stronger.

The UV–VIS spectra of the samples obtained in the presence of inorganic alkalis using both preparation methods, as depicted in Figure 3A,B, showed a large absorption band in the wavelength range of 240–380 nm, with maxima at 348 nm and 359 nm for the LDH. This shifting was due to the presence of zincite phase in higher amounts for the samples obtained through mechano-chemical method.

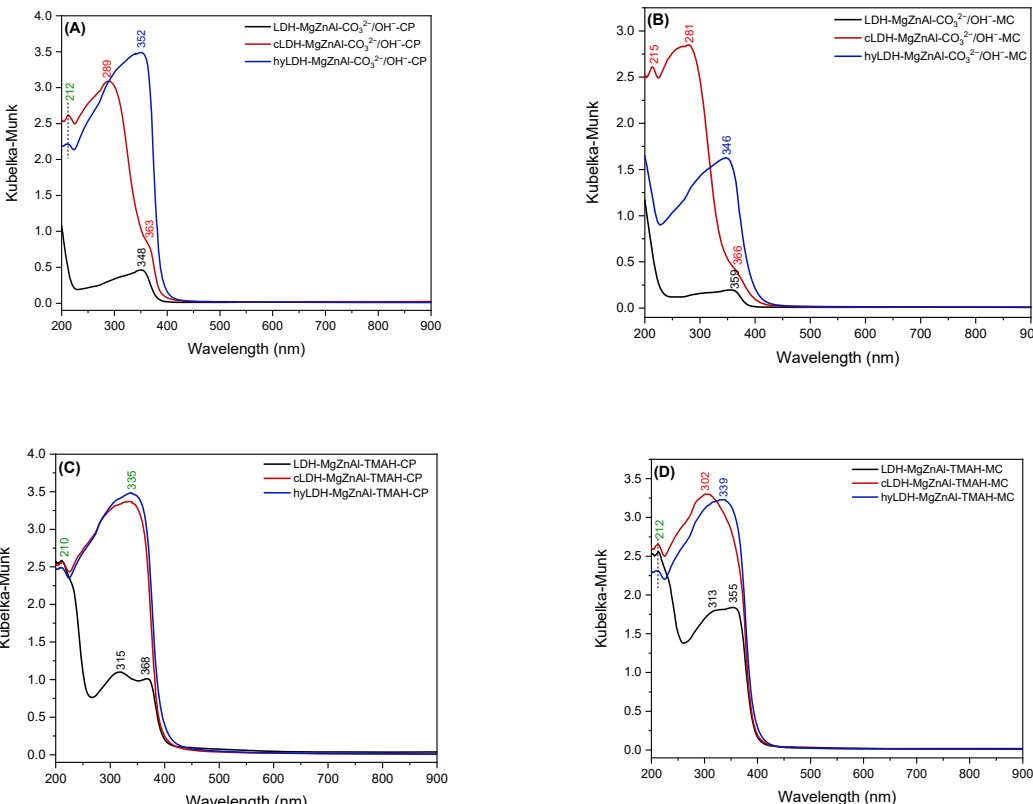

**Figure 3.** UV–VIS spectra of the materials synthesized though (**A**) co-precipitation and (**B**) mechano-chemical method, both in the presence of Na$_2$CO$_3$/NaOH, as well as (**C**) co-precipitation and (**D**) mechano-chemical method, both in the presence of TMAH.

In mixed oxide samples, the maxima identified at 363 nm and 366 nm corresponded to zinc oxide nanoparticles as impurities in the materials [46], while those at 289 nm and 281 nm were attributed to the band gap absorption in ZnO/MgO nanocomposites, which present larger band gaps as compared to ZnO [47]. The bands at 212 and 215 nm indicated the presence of $Mg(OH)_2$ and MgO [48]. For samples obtained in the presence of TMAH by both routes, as shown in Figure 3C,D, the common band at 210 nm indicated the presence of the organic base in their structure, while those at 313 nm and 315 nm were due to the presence of zincite phase inside the layered structure. Mixed oxides and reconstructed LDH also showed bands for zincite phase shifted to higher values for layered materials.

The basicity of samples, as shown in Table 2, decreased in the order mixed oxides > reconstructed LDH > parent LDH samples, a trend valid for both strong base sites as well as the sum of weak and medium base sites, regardless of the synthesis route. However, there was a slight increase in basicity for the samples prepared by mechano-chemical route. Additionally, the presence of small TMAH in LDH provided a pronounced base character compared to the LDH prepared with inorganic alkalis. In the meantime, the reconstructed LDH showed a decreased specific surface area comparative to those of the parent ones, due to the tendency of hexagonal lamellar crystals to cluster together in large conglomerate systems under vermiculate form, involving less defined platelets, which affect the access of reactants as well as the probe molecule $N_2$ to active sites [49,50]. However, the partial replacement of carbonate anions with hydroxyl groups following the calcination–hydration process leads to an increase in the basicity of the reconstructed samples due to the pronounced base character of these groups, as shown in Table 2. Regarding the textural properties of mixed oxides, they are in the proper range of these types of materials [4].

**Table 2.** The surface area and basicity of materials.

| Hydrotalcite Samples | Surface Area ($m^2 \cdot g^{-1}$) | Pore Volume ($cm^3 \cdot g^{-1}$) | Average Pore Width (Å) | Total Number of Base Sites ($mmol \cdot g^{-1}$) * | Distribution of Base Sites | |
|---|---|---|---|---|---|---|
| | | | | | Strong Base Sites ($mmol \cdot g^{-1}$) ** | Weak and Medium Base Sites ($mmol \cdot g^{-1}$) *** |
| LDH-MgZnAl-$CO_3^{2-}$/$OH^-$-CP | 69 | 0.387 | 222 | 7.23 | 0.48 | 6.75 |
| cLDH-MgZnAl-$CO_3^{2-}$/$OH^-$-CP | 258 | 0.843 | 124 | 10.38 | 0.53 | 9.85 |
| hyLDH-MgZnAl-$CO_3^{2-}$/$OH^-$-CP | 25 | 0.186 | 238 | 8.93 | 0.57 | 8.36 |
| LDH-MgZnAl-$CO_3^{2-}$/$OH^-$-MC | 150 | 0.498 | 132 | 7.42 | 0.51 | 6.91 |
| cLDH-MgZnAl-$CO_3^{2-}$/$OH^-$-MC | 266 | 0.845 | 121 | 10.48 | 0.57 | 9.91 |
| hyLDH-MgZnAl-$CO_3^{2-}$/$OH^-$-MC | 28 | 0.203 | 208 | 8.99 | 0.61 | 8.38 |
| LDH-MgZnAl-TMAH-CP | 2 | 0.005 | 115 | 7.71 | 0.50 | 7.21 |
| cLDH-MgZnAl-TMAH-CP | 235 | 0.765 | 112 | 10.64 | 0.55 | 10.09 |
| hyLDH-MgZnAl-TMAH-CP | 1 | 0.004 | 118 | 9.36 | 0.59 | 8.77 |
| LDH-MgZnAl-TMAH-MC | 44 | 0.118 | 106 | 7.85 | 0.54 | 7.31 |
| cLDH-MgZnAl-TMAH-MC | 241 | 0.798 | 124 | 10.66 | 0.56 | 10.10 |
| hyLDH-MgZnAl-TMAH-MC | 9 | 0.104 | 118 | 9.40 | 0.64 | 8.76 |

* mmol of acrylic acid. ** mmol of phenol. *** the difference between total number of base sites—strong base sites.

TMAH used as organic alkali in the synthesis of LDH by both the co-precipitation and mechano-chemical methods acted also as template molecule, as shown in Figure 4. The mechano-chemical method compared to the co-precipitation method in the presence of traditional inorganic alkali led to different pore widths (i.e., with maxima at 40 Å, 57 Å, and 150 Å, respectively). While preparation with traditional inorganic alkali led to a large pore width, with maxima of 358 Å for LDH-MgZnAl-$CO_3^{2-}$/$OH^-$-CP and 150 Å for LDH-MgZnAl-$CO_3^{2-}$/$OH^-$-MC, the preparation in the presence of organic alkali tended to decrease the pore width, leading to two well-defined maxima, as shown in Figure 4 C,D: one at 36 Å for LDH-MgZnAl-TMAH-CP and 34 Å for LDH-MgZnAl-TMAH-MC, and

a second one at 132 Å for LDH-MgZnAl-TMAH-CP and 90 Å for LDH-MgZnAl-TMAH-MC. The appearance of these two domains was not unusual, because the porosity was due to the size of the organic compounds, tri-methyl amine having the lower value and TMAH having the higher one.

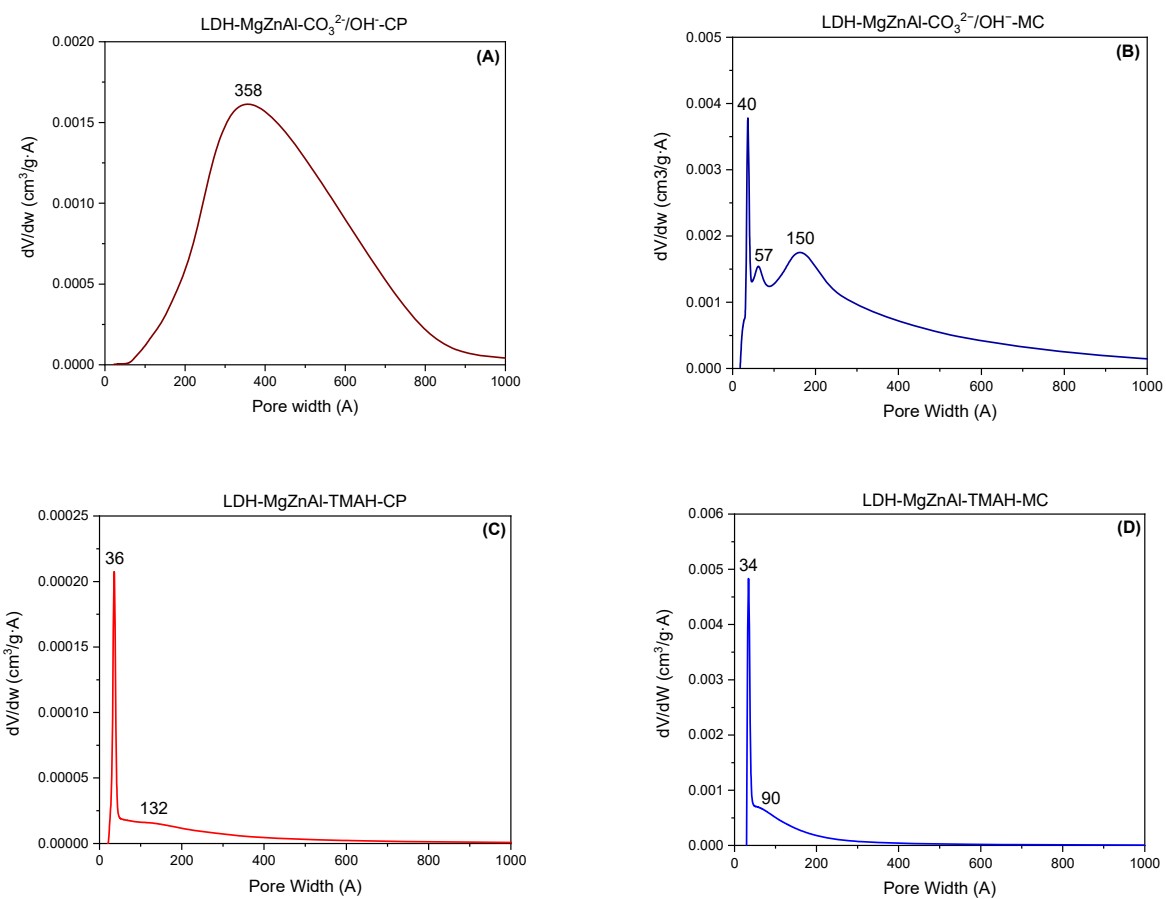

**Figure 4.** The representation of LDH sample pore widths; (A) LDH-MgZnAl-$CO_3^{2-}$/$OH^-$-CP; (B) LDH-MgZnAl-$CO_3^{2-}$/$OH^-$-MC; (C) LDH-MgZnAl-TMAH-CP; (D) LDH-MgZnAl-TMAH-MC.

### 2.2. Catalytic Activity

#### 2.2.1. Claisen–Schmidt Condensation

The blank test at room temperature after 5 h was in the range of experimental errors of 0.97% in cyclohexanone conversion, increasing at 9.13% at 120 °C and 2 h, with total selectivity toward 2,6-DBCHO (Scheme 1). The cyclohexanone conversion followed the same order: mixed oxides > reconstructed LDH > parent LDH samples, regardless of the preparation method, with total selectivity toward 2,6-DBCHO, as shown in Figure 5. Additionally, there was an improvement in the catalytic activities of materials prepared in the presence of organic alkalis compared to those prepared with inorganic alkalis, regardless of the preparation methods. However, a linear dependence between total basicity and conversion values was determined, as shown in Figure 6. Notably, at the end of the reactions, no products from self-cyclohexanone condensations or benzoic acid were found in the analyzed reaction mixtures. The total selectivity towards 2,6-DBCHO was also explained by the ability of 2-BCHO to adsorb onto active sites from pores with no steric hindrance, due to the non-planar shape of this molecule. On the same note, the presence of zincite phase to a different extent may also play a role in increasing the yield

of 2,6-DBCHO. Lower activity was noticed for the samples prepared with inorganic alkalis by mechano-chemical method. The comparison of the method types as well as of the hydrolysis agents used reveals the benefits presented by the use of organic instead of inorganic alkalis, but also of the advantages of co-precipitation over the mechano-chemical method.

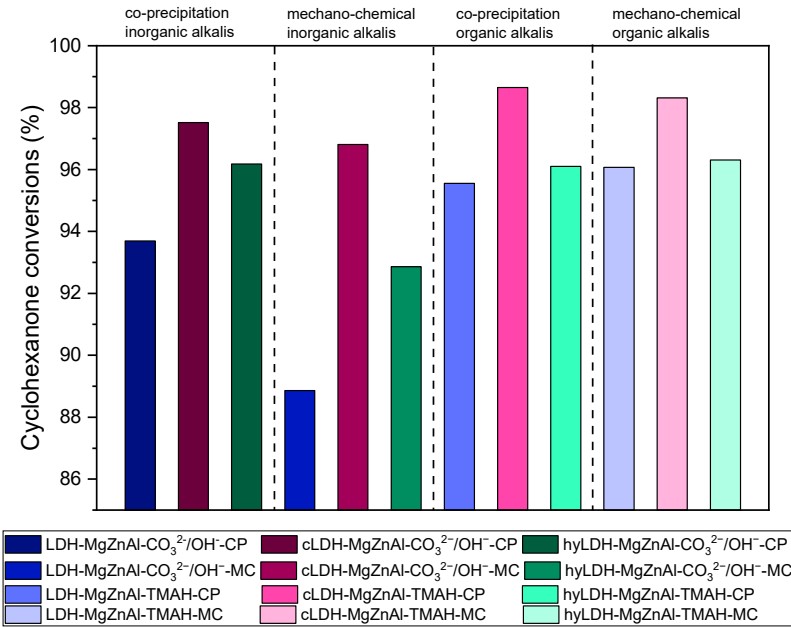

**Figure 5.** Cyclohexanone conversion after aldol condensation for 2 h, 120 °C, 20 mg of catalyst.

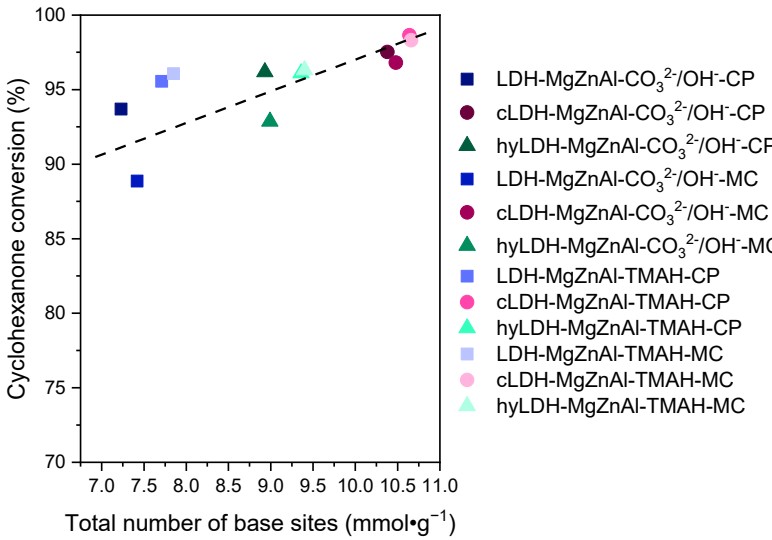

**Figure 6.** The cyclohexanone conversion vs. total number of base sites for investigated catalysts.

### 2.2.2. Cyclohexanone Self-Condensation

Blank reaction at room temperature or reflux after 5 h did not provide more than 0.19% of cyclohexanone conversion. The cyclohexanone conversions were significantly lower than those reached in Claisen–Schmidt condensation, and their magnitude follows

the same variation trend: mixed oxides > reconstructed LDH > parent LDH samples, regardless of the preparation method. There was also a high selectivity toward the *mono*-condensed product (Scheme 2), as shown in Table 3.

**Table 3.** Experimental data gathered for cyclohexanone conversion after 5 h, reflux, 5 wt.% catalyst, solvent-free.

| Catalysts | Conv. $C_6H_{10}O$ (%) | Sel. A (%) | Sel. A1 (%) | Sel. B (%) | Sel. B1 (%) |
|---|---|---|---|---|---|
| LDH-MgZnAl-CO$_3^{2-}$/OH$^-$-CP | 1.01 | 69.00 | 27.31 | 1.93 | 1.76 |
| cLDH-MgZnAl-CO$_3^{2-}$/OH$^-$-CP | 1.09 | 75.31 | 24.31 | 0.22 | 0.16 |
| hyLDH-MgZnAl-CO$_3^{2-}$/OH$^-$-CP | 0.70 | 69.74 | 29.64 | 0.44 | 0.17 |
| LDH-MgZnAl-CO$_3^{2-}$/OH$^-$-MC | 0.45 | 66.39 | 33.61 | 0.00 | 0.00 |
| cLDH-MgZnAl-CO$_3^{2-}$/OH$^-$-MC | 0.80 | 73.74 | 26.26 | 0.00 | 0.00 |
| hyLDH-MgZnAl-CO$_3^{2-}$/OH$^-$-MC | 0.49 | 69.87 | 30.13 | 0.00 | 0.00 |
| LDH-MgZnAl-TMAH-CP | 2.65 | 73.35 | 11.36 | 8.32 | 6.97 |
| cLDH-MgZnAl-TMAH-CP | 7.29 | 82.48 | 15.08 | 1.70 | 0.74 |
| hyLDH-MgZnAl-TMAH-CP | 6.23 | 75.73 | 12.20 | 1.53 | 0.55 |
| LDH-MgZnAl-TMAH-MC | 0.78 | 76.82 | 19.75 | 1.98 | 1.45 |
| cLDH-MgZnAl-TMAH-MC | 1.03 | 80.30 | 19.70 | 0.00 | 0.00 |
| hyLDH-MgZnAl-TMAH-MC | 0.67 | 74.21 | 25.79 | 0.00 | 0.00 |

This changing in paradigm compared to that presented for Claisen–Schmidt condensation, where the selectivity toward the *di*-condensate compound was total, was due to the rigidity of the double bond connecting cyclohexane moieties in the *mono*-condensate compound, which led to steric hindrances in accessing the porous structure of the catalyst. This fact was confirmed also by the dependence of cyclohexanone conversion on total number of base sites Figure 7.

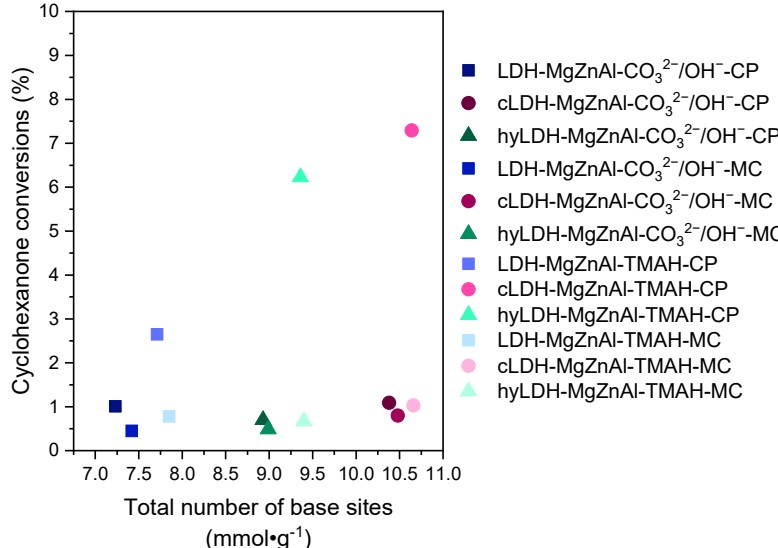

**Figure 7.** Cyclohexanone conversion vs. total number of base sites of investigated catalysts in self-cyclohexanone condensation.

The selectivities in the *di*-condensation product did not exceed 10%, as their occurrence was related only to the active sites on the external surface of the solid catalysts.

Therefore, the best catalytic activities were presented by the materials obtained in the presence of organic alkalis and the co-precipitation method.

### 2.3. Catalyst Reusability

The stability of catalysts (LDH-MgZnAl-TMAH-CP, cLDH-MgZnAl-TMAH-CP and hyLDH-MgZnAl-TMAH-CP) was checked in three consecutive Claisen–Schmidt condensation runs. After that, the conversion decreased to less than 4%, with no modification of diffraction lines in XRD patterns, as shown in Supplementary Materials, Figure S1, thus confirming the stability of these materials in this reaction, Figure 8.

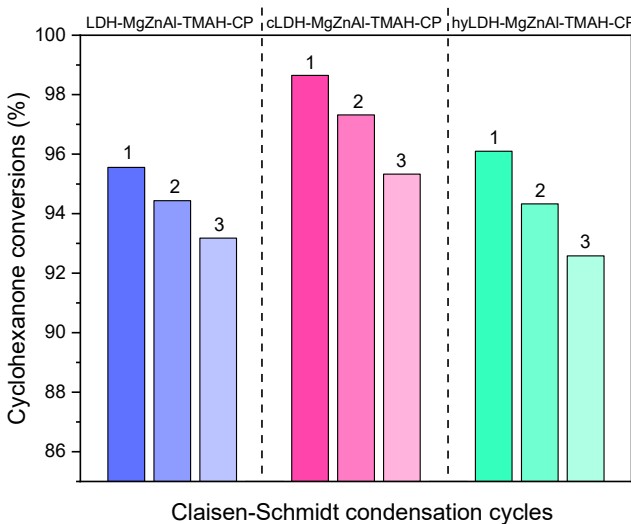

**Figure 8.** The catalyst reusability after 3 cycles in Claisen–Schmidt condensation for catalysts prepared by co-precipitation with organic alkalis.

## 3. Materials and Methods

### 3.1. Catalyst Preparation

The layered double hydroxide $Mg_{0.325}Zn_{0.325}Al_{0.25}$ was synthesized through both co-precipitation and the mechano-chemical method using the traditional inorganic alkalis, but also using a non-conventional organic base. Co-precipitation was carried out with inorganic alkalis according to a methodology already reported [44], mixing solutions of $Mg(NO_3)_2 \cdot 6H_2O$, $Zn(NO_3)_2 \cdot 6H_2O$, and $Al(NO_3)_3 \cdot 9H_2O$ (0.325/0.325/0.25 molar ratio and 1.5M) in the presence of NaOH and $Na_2CO_3$ ($NaOH/Na_2CO_3$ of 2.5 molar ratio and 1M $Na_2CO_3$) with a pH of 10, at room temperature and at a rate of 600 rpm. The TIM854, NB pH/EP/Stat pH-STAT Titrator was used to add both solutions at a feed rate of 60 mL·h⁻¹. The suspension was then aged for 18 h at 80 °C in an air atmosphere, cooled at room temperature and filtered, washed with bi-distilled water until a pH of 7 was attained, and dried for 24 h in air at 120 °C (**LDH-MgZnAl-CO₃²⁻/OH⁻-CP**). After that, calcination of this sample at 460 °C for 18 h in air led to the corresponding mixed oxide (**cLDH-MgZnAl-CO₃²⁻/OH⁻-CP**). The reconstruction of the layered structure was completed through memory effect by the immersion of cLDH-MgZnAl-CO₃²⁻/OH⁻-CP in bi-distilled water for 24 h at room temperature, followed by drying for 24 h at 120 °C in air (**hyLDH-MgZnAl-CO₃²⁻/OH⁻-CP**). The nontraditional mechano-chemical method was carried out by a direct milling of the precursors in a Mortar Grinder RM 200 for 1 h at 100 rpm (**LDH-MgZnAl-CO₃²⁻/OH⁻-MC**) at a pH of approx. 10, with no water addition or additional aging process. Further protocols for mixed oxides and reconstructed layered

samples were identical to the co-precipitation method (**cLDH-MgZnAl-CO₃²⁻/OH⁻-MC** and **hyLDH-MgZnAl-CO₃²⁻/OH⁻-MC**). Both the co-precipitation and mechano-chemical routes were also used to generate LDH in the presence of a nontraditional organic alkali represented by TMAH.

(TetraMethylAmonium Hydroxide; 25 wt.% in water) maintaining the same operational parameters (**LDH-MgZnAl-TMAH-CP**; **cLDH-MgZnAl-TMAH-CP**; **hyLDH-MgZnAl-TMAH-CP**). Notably, the bi-distilled water used in the washing step was 10 times lower compared to when inorganic alkalis were used as precipitation agents, where 2500 mL bi-distilled water was used to obtain 10 g of catalyst. The pH values during the co-precipitation and the washing stages were monitored using a Consort 853 Multiparameter equipped with pH electrode (Fisher Scientific, Turnhout, Belgium). A volume of TMAH identical to the one employed in the co-precipitation route was applied to obtain the LDH sample by mechano-chemical route (**LDH-MgZnAl-TMAH-MC**) under similar operating conditions (**cLDH-MgZnAl-TMAH-MC**; **hyLDH-MgZnAl-TMAH-MC**).

Index: LDH—layered double hydroxides; cLDH—mixed oxides; hyLDH—reconstructed layered structure; MgZnAl—the involved cations; CO₃²⁻/OH⁻—inorganic alkalis (Na₂CO₃/NaOH); TMAH—tetramethylammonium hydroxide; CP—co-precipitation; MC—mechano-chemical.

### *3.2. Catalyst Characterization*

Powder X-ray diffraction patterns were recorded using a Shimadzu XRD 7000 diffractometer with Cu K$\alpha$ radiation ($\lambda$ = 1.5418 Å, 40 kV, 40 mA) at a scanning speed of 0.10°·min⁻¹ in the 2θ range of 5–80°. DRIFTS spectra were recorded with JASCO FT/IR-4700 spectrometer by an accumulation of 128 scans in 400–4000 cm⁻¹ domain. DR UV–VIS spectra were recorded in the range 900–200 nm on a Jasco V-650 UV–VIS spectrophotometer with integration sphere, using Spectralon as a white reference. N₂ adsorption–desorption isotherms were determined using a Micromeritics ASAP 2010 instrument, where prior to nitrogen adsorption, samples were outgassed under vacuum for 24 h at 120 °C. The distribution of the pore sizes was determined by BJH desorption dV/dw pore volume using Faas correction. The original N₂ sorption isotherms are included in the Supplementary Materials, Figure S2. The base character of the catalysts was determined by an irreversible adsorption of organic molecules of different *pKa* method [51–53] (e.g. acrylic acid, *pKa* = 4.2, for total base sites and phenol, *pKa* = 9.9, for strong base sites), where the number of weak and medium base sites was calculated as the difference between the amounts of adsorbed acrylic acid and phenol.

### *3.3. Catalytic Tests*

### 3.3.1. Claisen–Schmidt Condensation

The Claisen–Schmidt condensation was carried out in a thermo-stated glass reactor with a water-cooled condenser, where a mixture of benzaldehyde (0.002 moles ReagentPlus >99%, Sigma-Aldrich, Darmstadt, Germany), cyclohexanone (0.001 moles >99%, Sigma-Aldrich) and 20 mg of catalyst (at a benzaldehyde/catalyst ratio of 10/1) was stirred under solvent-free conditions for 2 h at 120 °C [37]. After that the catalyst was removed by filtration, the mixture was washed with 1 mL of toluene and analyzed by Thermo-Quest GC with a FID detector and a capillary column 30 m in length with a DB5 stationary phase. The compounds were identified by mass spectrometer coupled chromatography, using a GC/MS/MS Varian Saturn 2100 T equipped with a CP-SIL 8 CB Low Bleed/MS column 30 m in length and 0.25 mm in diameter.

**Scheme 1.** Claisen–Schmidt condensation between benzaldehyde and cyclohexanone.

### 3.3.2. The Aldol Cyclohexanone Self-Condensation

The aldol cyclohexanone self-condensation was investigated in a similar reactor to the one utilized for the Claisen–Schmidt condensation, mixing 0.01 moles of cyclohexanone with 5 wt.% catalyst under solvent-free conditions [54]. After 5 h at reflux, the catalyst was removed from the mixture by filtration and the liquid reaction mixture was analyzed by GC-FID. Additionally, mass spectrometer coupled chromatography was used for compound identification.

**Scheme 2.** The aldol self-condensation of cyclohexanone; (A1) – 2-(1-cyclohexenyl)-cyclohexanone; (A) – 2 –cyclohexylidenecyclohexanone; (B) – 2,6-dicyclohexylidenecyclohexanone; (B1) – 2,6-di(1-cyclohexenyl)cyclohexanone.

### 3.4. Catalyst Recycling

The LDH-MgZnAl-TMAH-CP, cLDH-MgZnAl-TMAH-CP, and hyLDH-MgZnAl-TMAH-CP, the most active catalysts in their class, were selected for recycling tests. The layered catalysts were separated from the reaction mixture by filtration, washed with 1 mL of toluene and dried for 5 h at 120 °C in air before being used in the consecutive cycles. The same parameters were used for the mixed oxide with the specification that the thermal treatment was carried out at 460 °C.

### 4. Conclusions

Both preparation methods employed led to the obtaining of LDH materials spiked with zincite phase. However, the amount of additional phase was higher in mechanochemically prepared catalysts. All solids exhibited the memory effect of the Mg/Al LDH phase, while Zn was conserved as stable zincite phase. Using TMAH as an organic alkali for LDH preparation brings a number of advantages: *(i)* a smaller quantity of water in-

volved in the washing step, *(ii)* preventing contamination with alkali metal cations, and *(iii)* tailoring the LDH texture. Mechano-chemically prepared materials show a pronounced basicity compared than that of co-precipitated ones, while LDH materials prepared with TMAH present a higher basicity than that of LDH prepared with inorganic alkalis. Regardless of the preparation methods, organic/inorganic alkalis or reactions, the activity of the catalysts decreased in the order: mixed oxides > reconstructed LDH > parent LDH samples. For Claisen–Schmidt condensation, the conversions were higher than 90% after 2 h with a total selectivity toward 2,6-dibenzylidenecyclohexanone, while in the self-condensation of cyclohexanone, the conversion did not exceed 7.29% after 5 h. The yields reached for 2,6-dibenzylidenecyclohexanone after 2 h reaction were comparable to the highest yields reported by other authors who worked with more complex catalytic systems. These catalytic behaviors were a consequence of the cooperation between a number of active base sites, which promoted the activation of the reagent molecules and the shape selectivity due to the particular porous structure of the samples prepared with organic alkalis. The catalytic materials presented a good stability after three cycles in Claisen–Schmidt condensation.

**Supplementary Materials:** The following supporting information can be downloaded at: www.mdpi.com/article/10.3390/catal12070759/s1, Figure S1: XRD patterns of the catalysts recycled; Figure S2: Isotherm linear plot of LDH samples.

**Author Contributions:** Conceptualization, O.D.P.; Data curation, R.Z.; Funding acquisition, E.E.J.; Investigation, R.Z., S.-D.M. and O.D.P.; Methodology, B.C. and O.D.P.; Resources, M.T. and E.E.J.; Supervision, V.I.P. and S.O.; Writing—original draft, R.Z., O.D.P. and E.E.J.; Writing—review & editing, V.I.P., O.D.P. and S.O. All authors have read and agreed to the published version of the manuscript.

**Funding:** This work was financially supported by The Education, Scholarship, Apprenticeships and Youth Entrepreneurship Programmer—EEA Grants 2014-2021, Project No. 18-Cop-0041.

**Data Availability Statement:** The data are available on request from the corresponding author.

**Conflicts of Interest:** The authors declare no conflict of interest.

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
