# Peer review of "An Advanced Approach for MgZnAl-LDH Catalysts Synthesis Used in Claisen-Schmidt Condensation"

_catalysts, doi:10.3390/catal12070759_

Round 1

Reviewer 1 Report

In this manuscript, the authors reported the so-called advanced approach for MgZnAl-LDH catalysts synthesis towards Claisen-Schmidt condensation. In fact, it was not real new synthesis method for LDH. Anyway, the authors can improve it for publication in Catalysts after major revision.

      Special comments:

     (1) In Introduction, the authors  compared inorganic alkaline with organic one. Here, the authors should paid attention to COD pollution in washing water and treatment. Furthermore, generally organic alkaline is much higher price than inorganic one. On the other side, organic alkaline has been used to fabricate LDH, e.g., urea, hexamethylenetetramine.

(2) In Introduction section, the authors failed to summarize and clearly tell the potential readers which properties of the catalyst are important to this special reaction based on the references. Why and how to design and prepare this high-performace catalyst with MgZnAl-composition in this work?

(3) For Figures 1 and 2, the authors described too much and should pay attention to the difference caused by the different methods and explain the reasonable reasons.

(4) In my mind, the different methods may produce different morphologies and pore structure as well as surface properties, which are failed to show.  Even if on alkaline site, which alkaline site and what kind of alkaline site should be carefully analysized. All the properties should be related to the catalytic performance.

(5) The catalytic performance in the literature should be compared among the similar catalysts and those prepared in different ways in table.

(6) Refences should further up date.

Author Response

We thank both referees for their comments and recommendation. These have been a real help in improving the quality of the article.

(1) In Introduction, the authors compared inorganic alkaline with organic one. Here, the authors should paid attention to COD pollution in washing water and treatment. Furthermore, generally organic alkaline is much higher price than inorganic one. On the other side, organic alkaline has been used to fabricate LDH, e.g., urea, hexamethylenetetramine.

The use of urea and hexamethylenetetramine bring also the advantage of not using an extensive washing since they decompose into gases. Hence the same phenomenon occurs when using TMAH. Therefore the pollution of the washing water with COD is significantly decreased. From the point of view of the price according to Merck catalog the price of TMAH is slightly lower (27 euro/5 grams) than the price of hexamethylenetetramine (28 euro/5g), but the latter does not give the same effects in directing the porous structure as TMAH. Both urea and hexamethylenetetramine can act as precipitating agents by hydrolysis under pressure, while in the case of TMAH one can work at atmospheric pressure. The use of inorganic alkali, which are indeed cheaper, as precipitation agents requires higher amounts of water in the washing step, and practically cannot avoid the presence of Na traces in the resulting solids. Besides the waste washing waters have high alkalinity requiring an intensified water treatment process.

(2) In Introduction section, the authors failed to summarize and clearly tell the potential readers which properties of the catalyst are important to this special reaction based on the references. Why and how to design and prepare this high-performace catalyst with MgZnAl-composition in this work?

We have added some more explanations in the introduction of the revised manuscript.

(3) For Figures 1 and 2, the authors described too much and should pay attention to the difference caused by the different methods and explain the reasonable reasons.

We think that in this respect we have commented all that was necessary in order to highlight the effects of the organic alkali on the structure of the resulting solids. However we have reformatted the text in the discussion of the XRD in order to be seen more clearly and we have reformulated a sentence in the discussion of DRIFT.

(4) In my mind, the different methods may produce different morphologies and pore structure as well as surface properties, which are failed to show.  Even if on alkaline site, which alkaline site and what kind of alkaline site should be carefully analysized. All the properties should be related to the catalytic performance.

We have brought new data included in the Supplementary Material and we have also redrawn the Figure 4 based on the comments of reviewer 2.

(5) The catalytic performance in the literature should be compared among the similar catalysts and those prepared in different ways in table.

In the introduction section we have gathered all the performances obtained with the catalysts reported in literature for this reaction. We have highlighted that similar catalysts were not used. We have included a new sentence in the conclusion section.

(6) Refences should further up date.

We have included new reference.

Reviewer 2 Report

This manuscript reported the synthesis of LDH with TMAH as the base and tested the activity for Claisen-Schmidt condensation of the prepared materials in different forms. Some comments are attached below.

In Figure 4, the unit of the y-axis is ambiguous. If Figure 4 is the pore size distributions of different LDH samples, then the y-axis should be dV/dD, and the unit is cm3 g-1 A-1. The original N2 sorption isotherms need to be provided as well, and the method for deriving the pore size distribution should be added to the Materials and Methods section.

From lines 230-232, the authors mentioned that “the porosity is due to the size of the organic compounds, tri-methyl amine for the lower value and TMAH for the higher one”, which is not very clear and needs to be discussed further. From Figure 4, LDH-MgZnAl-TMAH-CP and LDH-MgZnAl-TMAH-MC both have well-defined and narrow small mesopores at 33 to 36 Angstroms, how are tri-methyl amine contributing to these pores? The relationship between the size of trimethyl amine molecules and the pore size is not very clear.

The data of the XRD patterns of the used catalysts (Line 289-291), can be added to the supporting information.

Since one of the advantages of using TMAH is that a smaller quantity of water is required in the washing step compared to NaOH/Na2CO3 as stated in the abstract and introduction, then it is important to provide more details in the experimental section. What procedure is applied for the washing of the solid? Was the washing performed in the Büchner funnel by soaking the solid in water, and for how long? Was a pH meter used to accurately monitor the pH? What is the volume of water used during the washing?

Authors claim that “Mechanochemically prepared materials show a pronounced basicity compared than that of the co-precipitated ones” in the conclusion section, although the data in Table 2 shows that mechanochemically prepared materials only have slightly higher basicity than co-precipitated ones. Were the adsorption experiments of acrylic acid and phenol performed multiple times to confirm the reproducibility? What are the standard deviations of the numbers of the base sites in Table 2?

In Line 391, “synergistic effect shown by the base sites” is not clearly shown and discussed in the main text, and the discussion section needs to be expanded to support this argument.

Author Response

(1) In Figure 4, the unit of the y-axis is ambiguous. If Figure 4 is the pore size distributions of different LDH samples, then the y-axis should be dV/dD, and the unit is cm3 g-1 A-1. The original N2 sorption isotherms need to be provided as well, and the method for deriving the pore size distribution should be added to the Materials and Methods section.

Thank you for this observation.  The figures were modified according to the suggestion. We have added in the Materials and Methods section the method for deriving the pore size distribution. The original N2 sorption isotherms were included in the supplementary information.

(2) From lines 230-232, the authors mentioned that “the porosity is due to the size of the organic compounds, tri-methyl amine for the lower value and TMAH for the higher one”, which is not very clear and needs to be discussed further. From Figure 4, LDH-MgZnAl-TMAH-CP and LDH-MgZnAl-TMAH-MC both have well-defined and narrow small mesopores at 33 to 36 Angstroms, how are tri-methyl amine contributing to these pores? The relationship between the size of trimethyl amine molecules and the pore size is not very clear.

Trimethyl amine is gaseous at room temperature and has a molecular size of 4.2 Å, while TMAH has molecular size of 5.593 Å. The evolution of TMA gas during the preparation is the factor contributing to the formation of small pores. The fact that the samples prepared by CP have a lower amount of small pores at 36 Å (0.00022) compared to the MC samples 34 Å (0.0049) indicates that during grinding a lot of TMAH is decomposed. Moreover during grinding an increased local heating is noticed as well compared to the CP.

(3) The data of the XRD patterns of the used catalysts (Line 289-291), can be added to the supporting information.

We have included these data in the Supplementary Materials

(4) Since one of the advantages of using TMAH is that a smaller quantity of water is required in the washing step compared to NaOH/Na2CO3 as stated in the abstract and introduction, then it is important to provide more details in the experimental section. What procedure is applied for the washing of the solid? Was the washing performed in the Büchner funnel by soaking the solid in water, and for how long? Was a pH meter used to accurately monitor the pH? What is the volume of water used during the washing?

We have added more information in the experimental section.

(5) Authors claim that “Mechanochemically prepared materials show a pronounced basicity compared than that of the co-precipitated ones” in the conclusion section, although the data in Table 2 shows that mechanochemically prepared materials only have slightly higher basicity than co-precipitated ones. Were the adsorption experiments of acrylic acid and phenol performed multiple times to confirm the reproducibility? What are the standard deviations of the numbers of the base sites in Table 2?

The adsorption experiments of acrylic acid and phenol were performed in triplicate. The standard deviations were +/- 2%, the values in the table are the average of the 3 determinations.

(6) In Line 391, “synergistic effect shown by the base sites” is not clearly shown and discussed in the main text, and the discussion section needs to be expanded to support this argument.

These catalytic behaviors are a consequence of the cooperation between the number of active base sites which promote the activation of the reagent molecules and the shape selectivity due to the particular porous structure of the samples prepared with organic alkali.

Round 2

Reviewer 1 Report

acceptable.

Reviewer 2 Report

The revised manuscript can be accepted in its current form.